# Effectiveness of ultrasound-guided percutaneous transhepatic biliary drainage to reduce radiation exposure: A single-center experience

**Sung Eun Park[1], In Chul Nam[2]\*, Hye Jin Baek[1], Kyeong Hwa Ryu[3], Sung Gong Lim[1], Jung Ho Won[4], Doo Ri Kim[2]**

1 Department of Radiology, Gyeongsang National University School of Medicine and Gyeongsang National University Changwon Hospital, Seongsan-gu, Changwon, Korea, 2 Department of Radiology, Jeju National University School of Medicine, Jeju Natuional University Hospital, Jeju, Korea, 3 Department of Radiology, Samsung Changwon Hospital, Changwon, Korea, 4 Department of Radiology, Gyeongsang National University College of Medicine and Gyeongsang National University Hospital, Jinju, Korea

\* sky_hall@naver.com

**Data Availability Statement:** All relevant data are within the paper and its Supporting Information files.

## Abstract

Percutaneous transhepatic biliary drainage (PTBD) has been an effective treatment to access the biliary tree, especially in case of endoscopically inaccessible biliary tree. In general, PTBD techniques are divided into two methods: fluoroscopy-guided PTBD and ultrasound (US)-guided PTBD. This study aimed to evaluate the effectiveness of US-guided PTBD, focusing on radiation exposure according to intrahepatic duct (IHD) dilatation degree, differences between right- and left-sided approaches and differences between benign and malignant biliary stenosis/obstruction. We evaluated technical success, clinical success, procedural data (the number of liver capsule punctures, procedural time, fluoroscopy time and radiation dose), and procedure-related complications. During the study period, a total of 123 patients with biliary stenosis/obstruction or bile leakage were initially eligible. We excluded 76 patients treated with only ERCP or initially treated with ERCP followed underwent PTBD insertion. Finally, a total of 50 procedures were performed in 47 patients. Of the 47 patients, 8 patients had anatomical alteration due to previous surgery, 6 patients refused ERCP, and 3 patients failed ERCP. For the remaining 30 patients, PTBD was performed on weekend or at night, 11 of whom had poor general condition, 10 patients underwent ERCP 3 to 4 days later after PTBD insertion, 6 patients improved after PTBD insertion without ERCP, 1 patient died, and 1 patient was referred to other hospital. Remaining 1 patient underwent surgery due to Mirizzi syndrome. All procedures were performed by two interventional radiologists. Technical success rate was 100%, clinical success was 94%, and the complication rate was 10%. Fluoroscopy time and the reported radiation dose were significantly lower in patients with dilated bile ducts than in those with non-dilated bile ducts, when biliary puncture under US guidance was performed initially. However, even in patients with non-dilated bile ducts undergoing initial trials of biliary puncture under US guidance, the fluoroscopy time and the reported radiation dose were low, based on current studies. No statistical significant differences were observed in terms of technical and dosimetry

**Funding:** The author(s) received no specific funding for this work.

**Competing interests:** The authors have declared that no competing interests exist.

results according to right-sided and left-sided procedures and benign and malignant biliary stenosis/obstruction. Thus, US-guided PTBD was found to be a safe and effective technique that significantly reduced fluoroscopy time and radiation doses.

## Introduction

Since the first report of fluoroscopy-guided percutaneous transhepatic biliary drainage (PTBD) in 1962 [1], the technique has been considered an effective treatment option for the treatment of benign or malignant bile duct stenoses or obstructions, especially in the case of endoscopically challenging access to the biliary tree [2]. Furthermore, due to the recent increase in the risk of post-surgical bile leak at the site of biliodigestive anastomosis, the number of PTBD procedures has also increased in daily practice in patients with non-dilated bile ducts [3–6]. Fluoroscopy-guided PTBD has been performed traditionally under anatomic landmarks with a high success rate [2]; however, biliary puncture per se is not one of the most difficult procedures but is associated with a potential risk of higher bleeding complications [7–10]. In addition, fluoroscopy-guided PTBD increases the exposure of patients and operators to radiation.

Currently, the technique of PTBD varies according to operator preferences and experience. The rapid advances in spatial and temporal resolution of ultrasound (US) provide increased access to the biliary tree under US guidance [7,11]. Several recent studies were conducted under US-guided PTBD. Giurazza et al. [12] reported that US guidance provides a safe and effective access the biliary tree for PTBD with low-grade complications and exposure to low radiation doses, based on multicenter experience. Lee et al. [13] reported successful PTBD in a sample of 50 patients with non-dilated bile ducts under combined US and fluoroscopic guidance. Until recently, however, investigations of radiation exposure occurred during PTBD procedure are little.

Therefore, the purpose of our study was to evaluate the safety and effectiveness of US-guided PTBD, especially focusing on fluoroscopy time and radiation dose according to the degree of intrahepatic duct (IHD) dilatation, differences between right- and left-sided approaches and differences between benign and malignant stenosis/obstruction.

## Materials and methods

### Patients

This Institutional Review Board of Gyeongsang National University Changwon Hospital approved the study (No.: GNUCH 2021-07-037). However, no patient approval or informed consent was required due to the retrospective nature of the study.

We retrospectively reviewed our institutional database from March 1, 2021 to July 1, 2021 and identified 123 patients with bile duct stenosis/obstruction or bile leakage. In general, the endoscopic team performed ERCP or alternative EUS-guided bile duct drainage as an initial treatment and bile duct decompression. Whereas in patients with altered anatomy due to previous surgery, patients who failed ERCP, who refused ERCP, patients with very poor general condition, or in case of weekends or nights, PTBD was inserted at first then ERCP was performed 3 to 4 days later. We excluded 76 patients treated by only ERCP or initially treated with ERCP followed underwent PTBD insertion, and the remaining 47 patients who underwent PTBD as an initial treatment were finally included. The case accrual process is

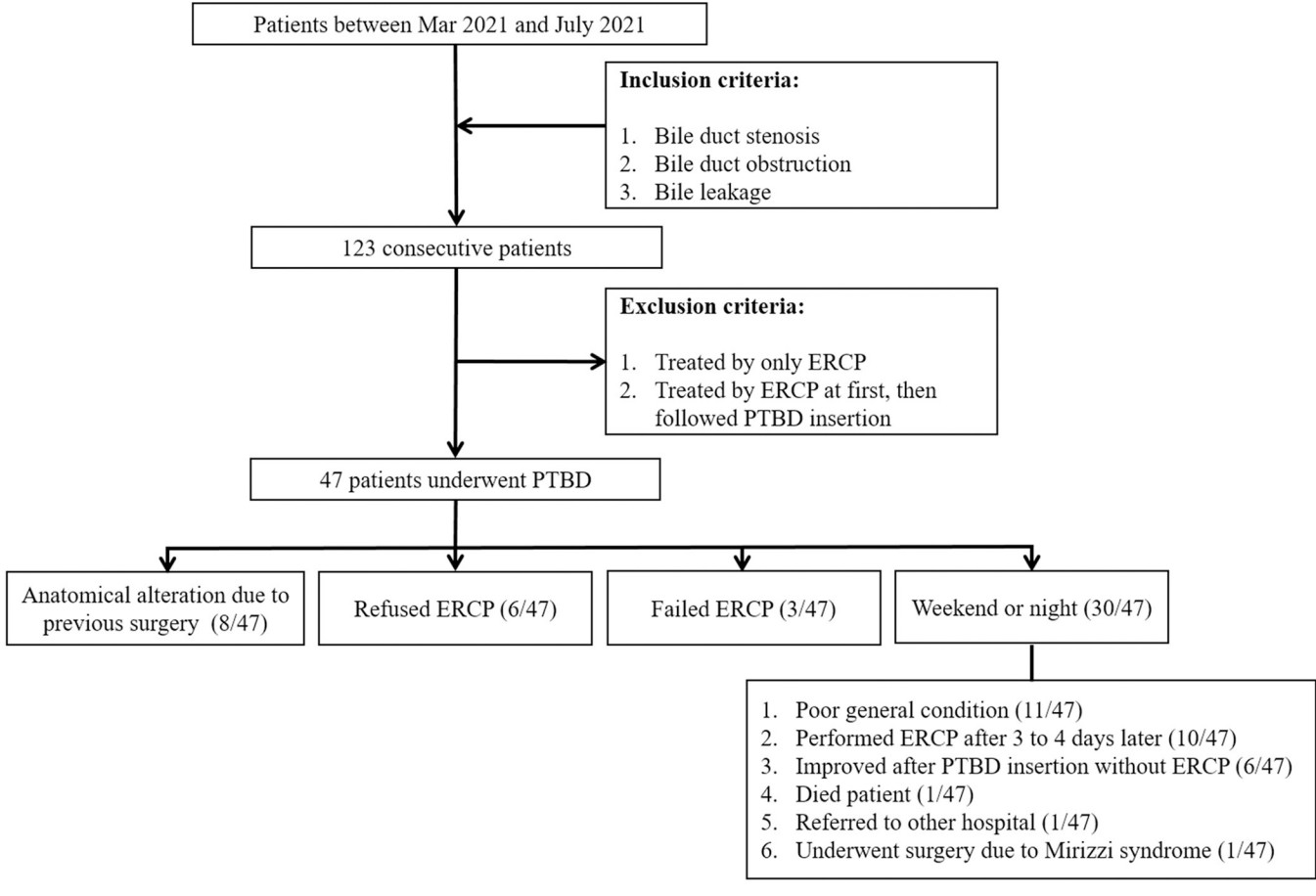

**Fig 1. Flow chart of the case accrual process.**

summarized in Fig 1. We recorded patients' medical history and data including age, gender, laboratory examination (aspartate transaminase (AST), alanine transaminase (ALT), gamma-glutamyl transferase (γ-GTP), bilirubin, white blood cell count, C-reactive protein) and reason for PTBD (benign or malignant stenosis/obstruction). We reviewed images on the picture archiving and communication system (PACS), and then divided patients into two groups according to the degree of IHD dilatation based on computed tomography (CT) or US findings at the time of the procedure. The groups were defined as follows: group I, non-dilated bile duct or minimally dilated bile duct, with invisible or barely visible IHD on CT or US; group II, dilated IHD, which is easily recognizable on CT or US, with at least half of the diameter of the portal veins. Technical success was defined as successful puncture of the IHD and subsequent positioning of a PTBD catheter in the biliary tree. Clinical success was defined as following criteria; 1) when the day of first inclusion of bilirubin value in a laboratory finding that was followed up after biliary drainage catheter insertion, 2) when bilirubin value significantly decreased compared to before PTBD insertion, resulting in no additional biliary drainage was clinically required. We also evaluated procedural data, including access procedures (right- or left-sided puncture), procedural time, number of liver capsule punctures (intended as passage of the needle through the hepatic capsule) to gain the biliary tree, total fluoroscopy time, and radiation dose. Complications were classified according to the Cardiovascular and Interventional Radiological Society of Europe (CIRSE) classification system [14]. Complications were considered to be clinically significant when they were graded as ≥3 on this scale.

## PTBD technique

All procedures were performed by two interventional radiologists with 2 and 5 years of experience in biliary intervention. The procedures were performed in an angio-suite (AlluraClarity FD20; Philips Healthcare). Before the procedure, liver US was performed to plan the procedure. Unless there was a specific reason, the approach (right- or left-sided) was decided upon by the operator.

Laboratory liver enzyme tests and coagulation status were routinely monitored according to the CIRSE guidelines for significant bleeding risk procedures [15]. Patients taking clopidogrel and aspirin were recommended to withhold for 5 days before the procedure. Transfusions were performed in the case of platelet values of <50,000 and the indexed normalized ratio (INR) was corrected if it was >1.5.

First, local anesthesia (10–15 ml of 2% lidocaine) was injected using a 22-gauge needle along the entrance point, inserted subcutaneously up to the liver capsule under US guidance and analgesia. A peripheral branch of the IHD was punctured with a 22G Chiba needle (Cook Medical) under US guidance with 3.5 MHz convex probe (Fig 2). In case of non-dilated IHD or barely visible IHD (as in group I), the peripheral branches of the portal vein were punctured (Fig 3). Next, the core needle was removed, and a mixture of iodine contrast and normal saline solution was carefully injected while withdrawing the needle until bile ducts were visible on fluoroscopy. In case of difficult IHD puncture despite multiple attempts under US guidance, the central duct was punctured under US or fluoroscopic guidance. The needle was left in place and a contrast medium was injected subsequently to opacify the peripheral IHD, which was then punctured under fluoroscopy guidance. In case of successful puncture of IHD, a 0.018″ nitinol micro-guidewire (Cook Medical) was advanced under fluoroscopic guidance, followed by insertion of a 4F introducer sheath (Neff Percutaneous Access Set, Cook Medical). Next, a 0.035″ hydrophilic guidewire (Terumo Corporation) was inserted into the distal common bile duct or small bowel. Finally, a PTBD drainage catheter (8.5 Fr/10 Fr) was inserted according to the operator's preferences and availability.

Based on as low as reasonably achievable (ALARA) principles, we significantly reduced the exposure to X-rays to the extent possible during procedure, using beam collimation, increasing the distance from patient to beam source, intermittent beam time, and avoidance of magnification.

## Statistical analysis

Student's *t* test or Chi squared analysis was used for comparison between group I and group II, between right- and left-sided approaches, and between benign and malignant biliary stenosis/ obstruction. The differences were considered statistically significant if the *p* value was <0.05. Continuous variables were expressed as the mean ± standard deviation (SD).

## Results

A total of 50 PTBD procedures involving 47 patients were reviewed. Of the 47 patients, 8 patients had anatomical alteration due to previous surgery (e.g., gastrectomy, etc.), 6 patients refused ERCP, and 3 patients failed ERCP. PTBD was performed on weekends or at night in 30 patients, 11 of whom had poor general condition (e.g., biliary sepsis, immediate postoperative state, etc.). 10 patients underwent ERCP after 3 to 4 day later, 6 patients improved after PTBD insertion without ERCP, 1 patient died, and 1 patient was referred to another hospital. Remaining 1 patient underwent surgery due to Mirizzi syndrome. The study patients consisted of 26 (55.3%) men and 21 (44.7%) women (age range, 33–95 years; mean age, 73.4 ± 13.3 years). Of the 47 patients, 20 patients (47%) underwent PTBD due to malignant obstructive

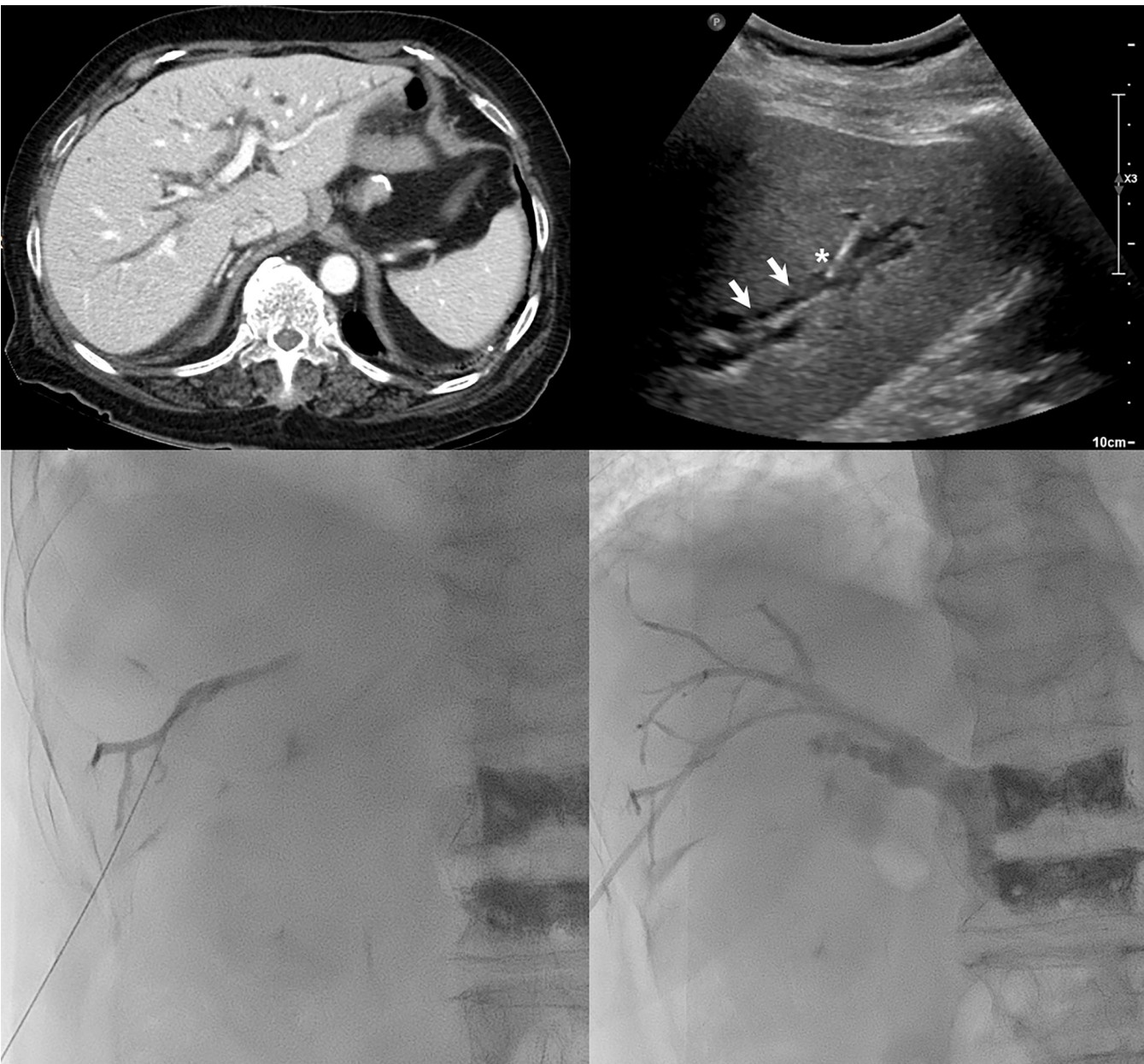

**Fig 2. An 83-year-old woman with jaundice due to distal cholangiocarcinoma.** A. Portal phase of contrast-enhanced computed tomography scan demonstrates diffuse dilatation of intrahepatic bile ducts. B. Intercostal right lobe ultrasonography scan shows diffuse dilatation of intrahepatic bile ducts (white arrow). Note that a 21-gauge Chiba needle was inserted into the target bile duct (white asterisk). C. Under fluoroscopy guidance, a mixture of contrast and normal saline was carefully injected and the bile duct confirmed. D. An 8.5 Fr drainage catheter was inserted at the distal common bile duct level.

jaundice, which included 10 patients with cholangiocarcinoma (50%), 3 patients with pancreatic head cancer (15%), and 2 patients diagnosed with gallbladder cancer (10%). The remaining 27 patients were subjected to PTBD due to benign obstructive jaundice, which included 18 cases of common bile duct stone with obstruction (66.6%), 7 cases of biliary sepsis (25.9%), and 2 cases of bile leakage after hepaticojejunostomy (7.4%). Disease associated with biliary drainage in 47 patients was summarized in Table 1.

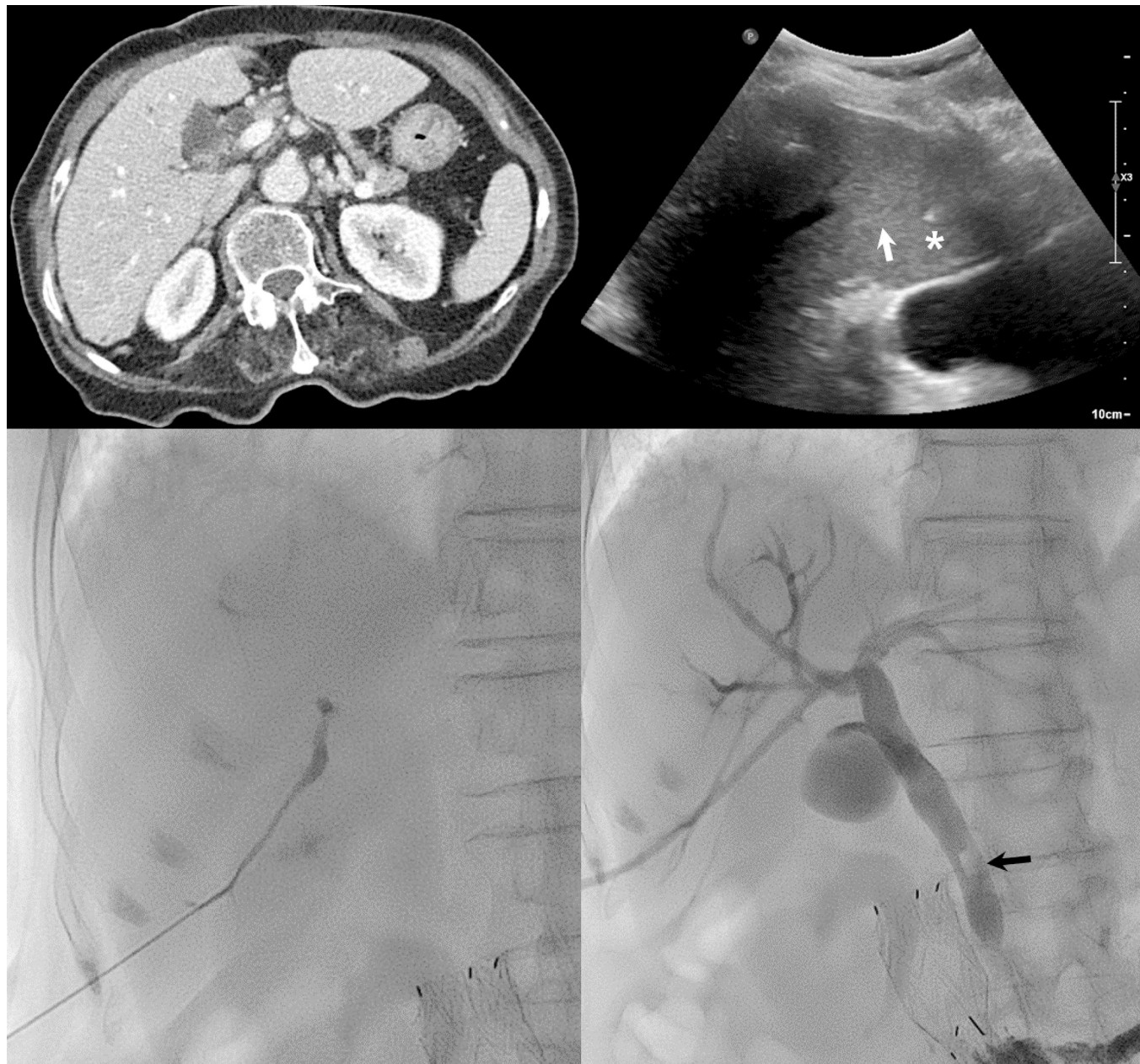

**Fig 3. An 86-year-old woman with jaundice due to common bile duct stone.** A. Venous contrast-enhanced computed tomography scan demonstrates minimal dilatation of intrahepatic bile ducts. B. Intercostal right lobe ultrasonography scan shows only portal vein (white arrow). A 21-gauge Chiba needle was inserted into the portal vein using a parallel technique (white asterisk). C. Insertion of the Chiba needle was followed by removal of the core needle, and a mixture of iodine contrast and normal saline solution was carefully injected while withdrawing the needle until bile ducts were visible on fluoroscopy. D. An 8.5 Fr drainage catheter was inserted at hilar level. Final cholangiography reveals a small stone in the distal common bile duct (black arrow).

The technical success, defined as successful puncture of the IHD and subsequent intubation of the biliary tree with a PTBD catheter, was 100%. The clinical success, defined as decrease in total bilirubin level after procedure, was 94%. The mean follow-up duration of bilirubin value was 4.1 ± 2.3 days (range, 1–8 days), and on overage, bilirubin value decreased to less than half of pre-PTBD.

**Table 1. Disease associated with biliary drainage in 47 patients.**

| Histopathology (n = 47) | Number of lesions (%) |
|---|---|
| Malignant lesions (n = 20) | |
| Cholangiocarcinoma | 10/20 (50%) |
| Pancreas head cancer | 3/20 (15%) |
| Gallbladder cancer | 2/20 (10%) |
| Peritoneal seeding metastasis | 1/20 (5%) |
| Lymphoma | 1/20 (5%) |
| Advanced gastric cancer | 1/20 (5%) |
| Ampulla of Vater cancer | 1/20 (5%) |
| Gastrointestinal stromal cancer | 1/20 (5%) |
| | |
| Benign lesions (n = 27) | |
| Common bile duct stone | 18/27 (66.7%) |
| Cholangitis | 7/27 (25.9%) |
| Bile leakage | 2/27 (7.4%) |

The mean total fluoroscopy time was 202.6 s ± 144.4 (range: 50–758). Concerning the X-ray dose assessment, the dose-area product (DAP) reported by the angiography machine was used. The mean total DAP was 5.2 ± 3.8 Gy cm$^2$ (range:1.4–19.4).

The overall mean number of liver capsule punctures performed to catheterize the biliary tree was 1.24 ± 0.56 (range: 1–3).

Of the 50 PTBD procedures, 15 procedures were considered as group I, and the remaining 35 procedures were included in group II. Laboratory values before and after procedure and demographics in groups I and II are summarized in Table 2. The mean number of liver capsule punctures was 1.73 ± 0.8 in group I and 1.03 ± 0.17 in group II ($P<0.001$). The mean procedure time was 12.3 ± 7.3 m in group I and 5.4 ± 1.8 m in group II ($P<0.001$). The mean fluoroscopy time was 317 ± 213.6 s (range: 50–758) in group I and 153.6 ± 54.9 s (range: 58–280) in group II ($P<0.001$). The mean DAP was 7.9 ± 4.1 Gy cm$^2$ (range: 2–19.4) in group I and 5.4 ± 1.9 Gy cm$^2$ (range: 1.4–9.5) in group II ($P<0.001$). Figs 4 and 5 present the box and whisker plots used to compare the fluoroscopy time and DAP of the two groups. In the case of the right-sided approach, a total of 37 procedures were performed, whereas the remaining 13 procedures were performed in the case of the left-sided approach. The mean number of liver capsule was 1.23 ± 0.6 in the case of the right-sided approach, 1.24 ± 0.55 in the case of the left-sided approach ($P = 0.945$). The mean procedure time was 7.4 ± 5 m in the right-sided approach and 7.4 ± 6.1 m in left-sided approach ($P = 0.953$). The mean fluoroscopy time was 201.3 ± 142.9 s (range: 50–758) in the right-sided approach, and 206.4 ± 154.6 s (range: 58–597) in the left-sided approach ($P = 0.914$). The mean DAP was 5.1 ± 3.9 Gy cm$^2$ (range: 1.4–19.4) in the right-sided approach, and 5.6 ± 3.2 Gy cm$^2$ (range: 2.2–12.7) in the left-sided approach ($P = 0.658$). In the case of benign stenosis/obstruction, 12 procedures were considered as group I, and the remaining 15 procedures were included in group II. Whereas in the case of malignant stenosis/obstruction, 3 procedures were considered as group I, and the remaining 20 procedures were in group II ($P = 0.009$). The mean number of liver capsule punctures was 1.13 ± 0.46 in the benign stenosis/obstruction, and was 1.33 ± 0.62 in the malignant stenosis/obstruction ($P = 0.201$). The mean procedural time was 8.3 ± 5.8 m in the benign stenosis/obstruction and 6.5 ± 4.8 m in the malignant stenosis/obstruction ($P = 0.237$). The mean fluoroscopy time was 223.2 ± 163.6 s (range: 82–758) in the benign stenosis/obstruction, and 178.4 ± 117 s (range: 50–597) in the malignant stenosis/obstruction ($P = 0.278$). The mean

**Table 2. Laboratory values before and after procedure and demographics in groups I and II.**

| | Group I (N = 15) | Group II (N = 35) | Statistical significance (*p* value) |
|---|---|---|---|
| Age | 73.1 ± 14.6 | 73.6 ± 12.4 | |
| Gender, *n* (%) | | | |
| Male | 9 (60) | 18 (51.4) | *P = 0.577* |
| Female | 6 (40) | 17 (48.6) | |
| Lesion | | | |
| Benign | 12 | 15 | *P = 0.01* |
| Malignancy | 2 | 18 | |
| CRP (mg/dL) | | | |
| Before | 97.3 ± 74.5 | 103.5 ± 76.4 | *P = 0.790* |
| After | 88.3 ± 67.4 | 71.9 ± 76.4 | *P = 0.555* |
| WBC (x10³/uL) | | | |
| Before | 12.2 ± 7.2 | 10.9 ± 5.5 | *P = 0.493* |
| After | 8.8 ± 5.2 | 8.7 ± 4.5 | *P = 0.959* |
| Bilirubin (mg/dL) | | | |
| Before | 2.9 ± 2.2 | 9.5 ± 8.5 | *P = 0.005* |
| After | 1.4 ± 0.9 | 4.6 ± 5.3 | *P = 0.028* |
| Follow-up duration (days) | 3.3 ± 2.3 | 4.5 ± 2.3 | *P = 0.065* |
| AST (IU/L) | | | |
| Before | 165.5 ± 132.4 | 170.7 ± 168.6 | *P = 0.917* |
| After | 69.9 ± 82.2 | 75.1 ± 63.5 | *P = 0.816* |
| ALT (IU/L) | | | |
| Before | 151.9 ± 134.4 | 141.8 ± 164 | *P = 0.836* |
| After | 68.6 ± 66.2 | 54.5 ± 58.3 | *P = 0.463* |
| γ-GTP | | | |
| Before | 399.3 ± 355.3 | 432.8 ± 387.6 | *P = 0.775* |
| After | 212.5 ± 192.1 | 248.1 ± 212.3 | *P = 0.579* |

CRP, C-reactive Protein; WBC, white blood cell; AST, asparate aminotransferase; ALT, alanine aminotransferase; γ-GTP, gamma-glutamyl transpeptidase.

DAP was 5.7 ± 4.4 Gy cm$^2$ (range: 2–19.4) in the benign stenosis/obstruction, and was 4.7 ± 2.8 Gy cm$^2$ (range: 1.8–12.6) in the malignant stenosis/obstruction ($P = 0.507$).

Complications occurred in five procedures (10%), including two cases of minor bleeding that was self-limited without further management, two cases of mild fever that was self-limited without further management, and one of a small subcapsular biloma that was detected at follow-up US, requiring antibiotic treatment for 2 weeks. The overall procedural data reported for group I, II, right-sided, left-sided approaches, benign and malignant stenosis/obstruction were summarized in Tables 3–5.

## Discussion

PTBD is an effective treatment option for malignant or benign obstructive jaundice, especially in case of endoscopically inaccessible biliary tree or bile leak [2–6]. The procedure consisted of cannulation of a peripheral bile duct after puncture followed by image-guided wire and catheter insertion. The indications for PTBD include decompression of biliary tree in acute illness, dilatation of stenosis or occlusion of bile duct using a balloon or stent, removal of biliary stones, biopsy for biliary tree, and biliary diversion during bile leak [16–18]. Several techniques have been proposed since the first description of PTBD under fluoroscopic guidance, however,

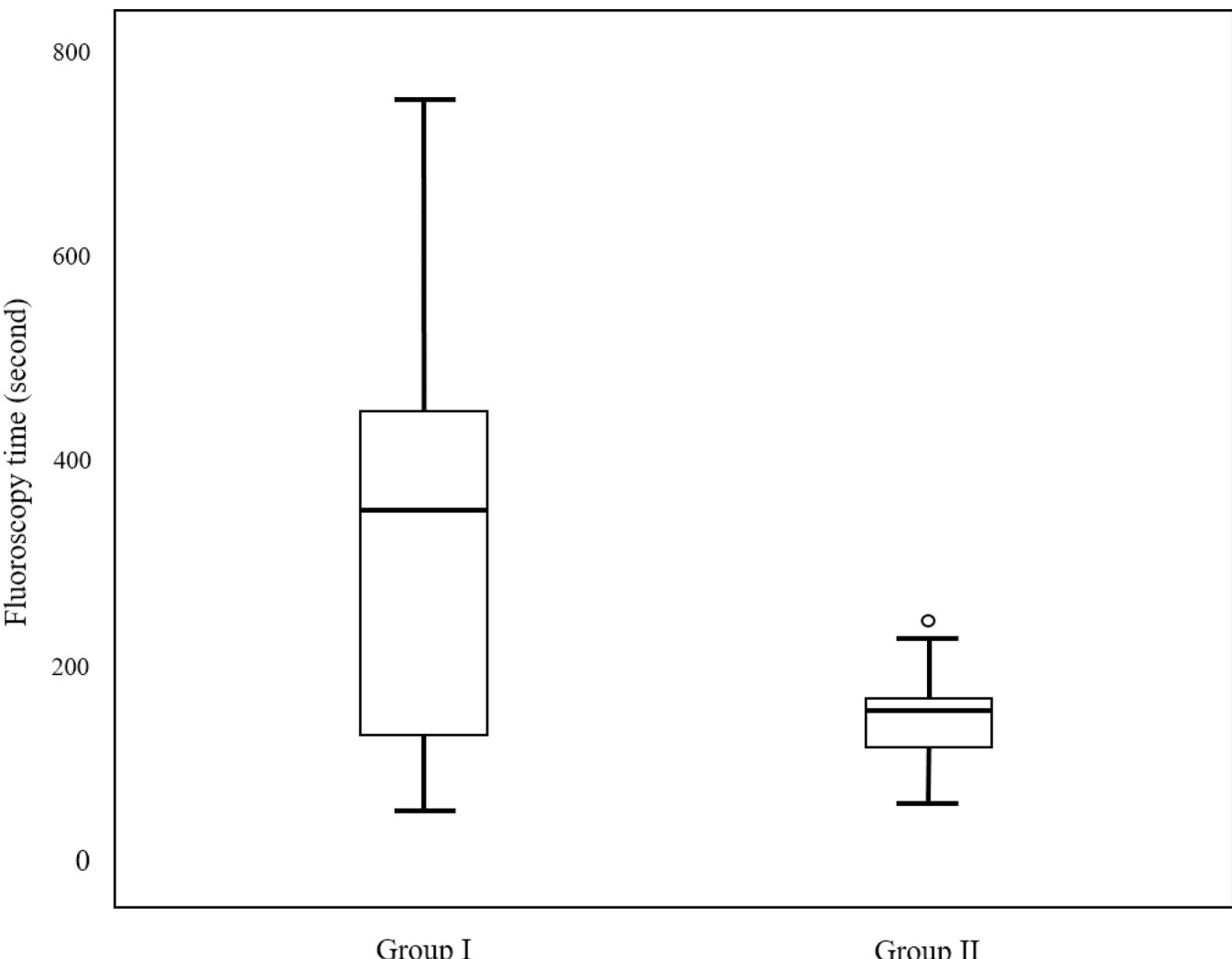

**Fig 4. A box and whisker plot comparing the fluoroscopy times between groups I and II.** The mean fluoroscopy time was 317 ± 213.6 s in group I and 153.6 ± 54.9 s in group II ($p$ = 0.01).

until recently, no clear global standard for PTBD was available. Multiple fluoroscopic and US-guided approaches have been reported for biliary puncture as a first, but only limited data support US-guided approach [7,11–13].

In this study, PTBD under US guidance was effective and safe for biliary drainage with a technical success rate of 100%, clinical success rate of 94%, and a minor complication rate of 10%. The results showed good agreement with the expected success rates defined by the CIRSE guidelines [14]. Also, the results are similar to recent reported studies [12,19].

In our study, the total mean fluoroscopy time and radiation dose were definitely lower when compared with recent studies [12,19–22]. Numerically, the 75th percentile of the distribution of median values of a dose acquired is defined as the diagnostic reference level (DRL) to promote optimal radiation protection of patients. Schegerer et al. [21] conducted a prospective study involving 16 hospitals from 13 countries, and reported that the 75th percentile of median value of DAP of PTBD was 23 Gy cm$^2$ and the 75th percentile of median value of fluoroscopy time was 600 s. Pedersoli et al. [19] reported that the mean fluoroscopy time was

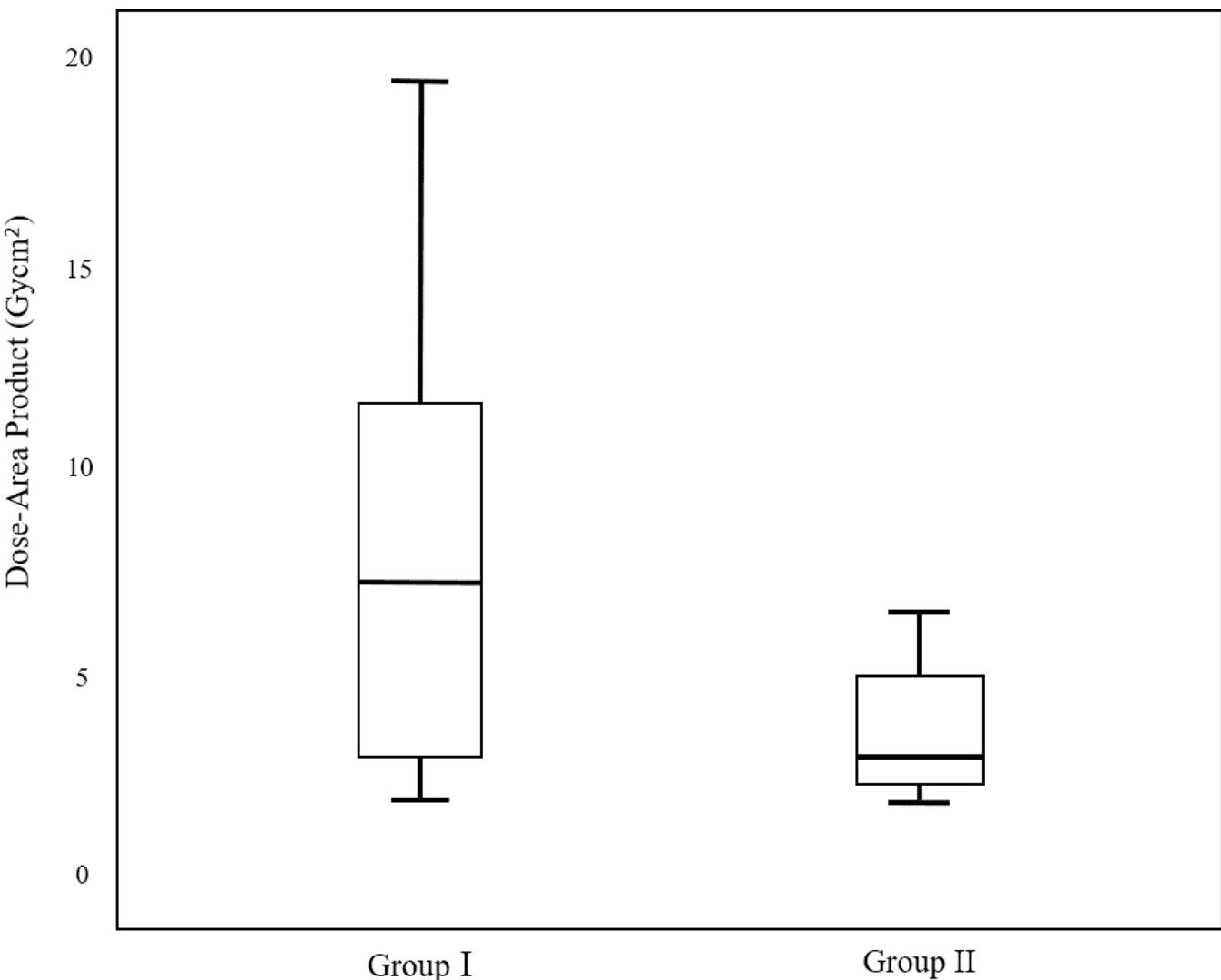

**Fig 5. A box and whisker plot comparing the dose-area product (DAP) of groups I and II.** The mean DAP was 7.9 ± 4.1 Gy cm$^2$ in group I and 5.4 ± 1.9 Gy cm$^2$ in group II ($p$ = 0.016).

**Table 3. Procedural data reported for groups I and II.**

| | Group I (N = 15) | Group II (N = 35) | Statistical significance (*p* value) |
|---|---|---|---|
| Clinical success (%) | 93.3 | 94.3 | $P$ = 0.896 |
| The No. of liver capsule punctures | 1.73 ± 0.8 | 1.03 ± 0.17 | $P<0.001$ |
| Procedure time (m) | 12.3 ± 7.3 | 5.4 ± 1.8 | $P<0.001$ |
| FT (s) | 317 ± 213.6 | 153.6 ± 54.9 | $P<0.001$ |
| DAP (Gy cm$^2$) | 7.9 ± 4.1 | 5.4 ± 1.9 | $P<0.001$ |
| Complications (%) | 13.3 | 8.6 | $P$ = 0.607 |

FT, fluoroscopy time; DAP, dose-area product.

**Table 4. Procedural data reported for right-sided and left-sided approaches.**

|  | Right (N = 37) | Left (N = 13) | Statistical significance (*p* value) |
|---|---|---|---|
| Clinical success (%) | 92 | 100 | P = 0.289 |
| The No. of liver capsule punctures | 1.23 ± 0.6 | 1.24 ± 0.55 | P = 0.945 |
| Procedure time (m) | 7.4 ± 5 | 7.4 ± 6.1 | P = 0.953 |
| FT (s) | 201.3 ± 142.9 | 206.4 ± 154.6 | P = 0.914 |
| DAP (Gy cm$^2$) | 5.1 ± 3.9 | 5.6 ± 3.2 | P = 0.658 |
| Complications (%) | 10.8 | 7.6 | P = 0.747 |

FT, fluoroscopy time; DAP, dose-area product.

00:42:36 ± 00:35:39 h in nondilated bile ducts and 00:30:28 ± 00:25:10 h in dilated bile ducts; the mean radiation dose was 18651 ± 17689 cGy cm$^2$ for non-dilated bile ducts and 14670 ± 16099 cGy cm$^2$ for dilated bile ducts. Giurazza et al. [12] reported that the mean total fluoroscopy time was 570.4 s and the mean DAP was 37.25 Gy cm$^2$ based on their experience. Lee et al. [22] reviewed nationwide DRLs for interventional procedures, especially involving PTBD, and reported a mean DAP 43 Gy cm$^2$. The mean fluoroscopy time was 849 s in the UK in 2012, and the mean DAP was 100 Gy cm$^2$. The mean fluoroscopy time was 1800 s in the US in 2012, and the mean DAP was 30.8 Gy cm$^2$. The mean fluoroscopy time was 1038 s in Spain in 2016, and the mean DAP was 36.1 Gy cm$^2$ and 427.5 s in South Korea in 2012. In Comparison, the fluoroscopy time and radiation dose were significantly lower in our study. These discrepancies in fluoroscopy time and DAP compared with other studies are encouraging and probably attributed to active use of US in initial biliary puncture and active compliance with ALARA principle.

We observed that fluoroscopy time and DAP were significantly lower in group II than in group I, as expected. The results suggest that fluoroscopy time and radiation dose can be significantly lowered by performing US immediately before the procedure and biliary puncture under US guidance initially in patients with dilated bile duct. Giurazza et al. [12] reported 100% technical success of US-guided PTBD in 117 patients in a multicenter study. In their case series, 13.3% cases showed non-dilated IHD; the mean total fluoroscopy time was 570.4 s and the mean DAP was 37.25 Gy cm$^2$. In our study, 30% cases showed non-dilated IHD; the mean total fluoroscopy time was 202.6 s and the mean DAP was 5.2 Gy cm$^2$. We achieved not only a 100% technical success rate of PTBD under combined US and fluoroscopic guidance, but also remarkably with a lower fluoroscopy time and radiation dose. This discrepancy appeared to be due to an appropriate combination of US and fluoroscopy guidance to approach biliary puncture when US-guided puncture was not feasible immediately.

**Table 5. Procedural data reported for benign and malignant biliary stenosis/obstruction.**

|  | Benign (N = 27) | Malignant (N = 23) | Statistical significance (*p* value) |
|---|---|---|---|
| Clinical success (%) | 92 | 96 | P = 0.458 |
| The No. of liver capsule punctures | 1.13 ± 0.46 | 1.33 ± 0.62 | P = 0.201 |
| Procedure time (m) | 8.3 ± 5.8 | 6.5 ± 4.8 | P = 0.237 |
| FT (s) | 223.2 ± 163.6 | 178.4 ± 117 | P = 0.278 |
| DAP (Gy cm$^2$) | 5.7 ± 4.4 | 4.7 ± 2.8 | P = 0.338 |
| Complications (%) | 7.4 | 13.1 | P = 0.507 |

FT, fluoroscopy time; DAP, dose-area product.

Even in patients with non-dilated bile ducts undergoing an initial trial of biliary puncture under US guidance, the fluoroscopy time and the reported radiation dose were low, based on current study [19]. Pedersoli et al. [19] reported that the mean fluoroscopy time was 00:42:36 ± 00:35:39 h and the mean radiation dose was 18.6 ± 17.6 Gy cm$^2$ for non-dilated bile ducts. Whereas in our study, the mean fluoroscopy time was 317 ± 213.6 s and the mean radiation dose was 7.9 ± 4.1 Gy cm$^2$ for non-dilated bile ducts. Even if compared with procedural time in our study, the mean procedural time was 12.3 ±7.3 m in nondilated bile ducts. Thus, US-guided PTBD was found to be a safe and effective technique significantly reducing fluoroscopy time and radiation dose.

There were no significant differences in the number of liver capsule punctures, procedural time, fluoroscopy time, DAP, and complication rates between right- and left-sided PTBD. Our results are consistent with recent studies [12,23]. Rivera-Sanfeliz et al. [23] reported a higher incidence of hemobilia following left- versus right-sided PTBD in their study, but the increased incidence did not reach statistical significance (*P* = 0.077). Giurazza et al. [12] also reported no significant differences in terms of procedural data or complications between right- and left-sided biliary procedures. However, Choi et al. [24] reported that left-sided PTBD was the only independent risk factor associated with hepatic arterial injury, and right-sided PTBD was preferable unless technical difficulty or secondary intervention warranted left-sided PTBD. Although our study findings were not able to conclude superiority for right- versus left-sided approaches, several studies have already discussed this issue and reported clear advantages/disadvantage for right- and left-side approaches. Chandrashekhara et al. [25] reported that the right-sided approach has the merits of less radiation exposure to the hands of performers and improvement of drainage stability relative to the left-sided approach, whereas the left-sided approach has the merits of being relatively easier to perform and better patients' compliance due to comfort improvement. Thus, we suggest that the approach of PTBD should be decided according to operator preferences and experience, and the clinical symptoms of the patient.

In cases with benign biliary stenosis/obstruction, the IHD was not markedly dilated compared with malignant biliary stenosis/obstruction. This results are well consistent with previous study [26]. Procedural time, fluoroscopy time and DAP in cases with malignant biliary stenosis/obstruction tended to be lower than those in cases with benign biliary stenosis/obstruction, however, there were no statistical significance. In addition, the number of liver capsule punctures was higher in group I than in group II with statistically significance, meaning the number of liver capsule puncture was correlated with IHD dilatation. These results suggested that the most difficult and time-consuming reason for the operator performing PTBD was directly related to the degree of IHD dilatation, rather than the reason for PTBD insertion (benign or malignant stenosis/obstruction) nor the approach of PTBD insertion (right- or left-sided).

This study's limitations should be considered when interpreting the results. First, this study was retrospectively designed, which suggests selection bias. Second, the sample size was small. Third, there was no direct comparison in terms of fluoroscopy time and radiation dose between fluoroscopy-guided PTBD and US-guided PTBD, because all biliary punctures were routinely performed under US guidance initially.

## Conclusion

In conclusion, we found that biliary puncture under US guidance was safe and effective and associated with low rates of complication and high rate of success. Fluoroscopy time and the reported radiation dose were significantly lower in patients with dilated bile ducts than in those carrying non-dilated bile ducts, when biliary puncture under US guidance was

performed initially. However, even in patients with non-dilated bile ducts undergoing an initial trial of biliary puncture under US guidance, the fluoroscopy time and the reported radiation dose were low, based on current studies. Thus, US-guided PTBD was found to be a safe and effective technique significantly reducing radiation exposure.

## Supporting information

**S1 File.**
(XLSX)

## Author Contributions

**Conceptualization:** In Chul Nam.

**Formal analysis:** In Chul Nam, Hye Jin Baek, Kyeong Hwa Ryu, Sung Gong Lim.

**Investigation:** In Chul Nam, Hye Jin Baek, Kyeong Hwa Ryu, Sung Gong Lim.

**Methodology:** Sung Eun Park, In Chul Nam.

**Supervision:** In Chul Nam.

**Writing – original draft:** Sung Eun Park, In Chul Nam.

**Writing – review & editing:** Sung Eun Park, In Chul Nam, Hye Jin Baek, Kyeong Hwa Ryu, Sung Gong Lim, Jung Ho Won, Doo Ri Kim.

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
