## [Decision Letter · Decision Letter 0]

13 Jul 2022

PONE-D-22-13247Effectiveness of Ultrasound-guided Percutaneous Transhepatic Biliary Drainage to Reduce Radiation Exposure: A Single-Center ExperiencePLOS ONE

Dear Dr. Nam,

Thank you for submitting your manuscript to PLOS ONE. After careful consideration, we feel that it has merit but does not fully meet PLOS ONE’s publication criteria as it currently stands. Therefore, we invite you to submit a revised version of the manuscript that addresses the points raised during the review process.

 You will find that the reviewers feel that your manuscript provides relevant data, but they also address open questions and limitations of your study. Please thoroughly revise your manuscript and provide requested information, especially on contraindication to ERCP treatment and the impact of anticoagulation/anti-platlet medication.

We look forward to receiving your revised manuscript.

Kind regards,

Thomas Karlas, M.D.

Academic Editor

PLOS ONE

Journal Requirements:

2. We note you have included a table to which you do not refer in the text of your manuscript. Please ensure that you refer to Tables 1 and 3 in your text; if accepted, production will need this reference to link the reader to the Table.

Reviewers' comments:

Reviewer's Responses to Questions

**Comments to the Author**

1. Is the manuscript technically sound, and do the data support the conclusions?

Reviewer #1: Yes

Reviewer #2: Yes

Reviewer #3: Partly

2. Has the statistical analysis been performed appropriately and rigorously? 

Reviewer #1: Yes

Reviewer #2: No

Reviewer #3: No

3. Have the authors made all data underlying the findings in their manuscript fully available?

Reviewer #1: Yes

Reviewer #2: Yes

Reviewer #3: Yes

4. Is the manuscript presented in an intelligible fashion and written in standard English?

Reviewer #1: Yes

Reviewer #2: Yes

Reviewer #3: Yes

5. Review Comments to the Author

Reviewer #1: This study evaluated the usefulness of US-guided PTBD in terms of reducing fluoroscopy time and radiation doses. Although the retrospective design without a comparison group is a major limitation of this study, the findings of this study is clinically significant to reduce radiation exposures.

1. Please add the exclusion criteria for PTBD in the Methods section.

2. This study included many cases of common bile duct stone cases. In cases with benign biliary stricture, the intrahepatic biliary tract is not markedly dilated compared with malignant biliary stenosis. I suggest the authors to compare fluoroscopy times between benign and malignant biliary strictures.

3. Is there a correlation between the number of punctures the punctured bile duct diameter?

Reviewer #2: The authors submitted a manuscript of a small retrospective, single-center study comparing the effectiveness of ultrasound-guided PTBD by radiologists in a small series of 47 patients. While I congratulate the authors for their hard work collating the data and the excellent success rate of 100% of their PTBDs I am not convinced of the novelty and impact of this investigation. In my eyes, the manuscript represents a case series with a detailed description. However, the descriptions are helpful for clinical routine and the manuscript is clear and well written. I made a few suggestions and I am happy to review the manuscript again should the authors decide to resubmit it.

Major

- In contrast to the presentation in the introduction, PTBD is not considered the gold standard for bile duct stenosis or obstructions. The authors should clarify why ERCP was not feasible in the series of 47 patients and why the alternative of EUS-guided bile duct drainage was not sought.

- The authors describe clearly their best practice how they perform PTBD with the excellent success rate of 100% even in non-dilated ducts. A few more details would be desirable: (i) Were antiplatet agents such as clopidogrel and aspirin also paused in patients with a clear indication (e.g. recent coronary stent)? (ii) was the puncture performed in a guided manner (e.g., ultrasound probe with preformed holes) or “free hand”)? (iii) what was the pulse rate for fluoroscopy

- How was “clinical success” exactly defined – i.e., in which interval should the total bilirubin level decrease and to which extent of the level pre-PTBD?

- The timepoint of laboratory controls should be clearly stated (e.g., 2 days after the procedure)

- Please show exact p values

Minor

- The abstract should clarify why ERCP was not initially feasible and that all PTBDs were performed by radiologists

- Please state the amount of punctures until successful bile duct cannulation was achieved. Was it higher in non-dilated bile ducts?

- Please clarify why PTBD should be better than ERCP for bile leak.

Reviewer #3: This study aimed to evaluate the effectiveness of US-guided PTBD (Percutaneous transhepatic biliary drainage), focusing on fluoroscopy time and radiation dose according to intrahepatic duct dilatation (IHD) degree, and differences between right- and left-sided approaches. The authors evaluated technical success, clinical success, procedural data (fluoroscopy time and radiation dose), and procedure-related complications. US-guided PTBD was found to be a safe and effective technique significantly reducing fluoroscopy time and radiation dose.

This study has some limitations:

1. it was retrospectively designed, which suggests selection bias.

2. the sample size was small.

3. there was no direct comparison in terms of fluoroscopy time and radiation dose between fluoroscopy-guided PTBD and US-guided PTBD, because all biliary punctures were routinely performed under US guidance initially.

Please focus more in the discussion section on the original elements of the study and on what you think this study brings in addition to what is already known.

6. PLOS authors have the option to publish the peer review history of their article (what does this mean?). If published, this will include your full peer review and any attached files.

Reviewer #1: No

Reviewer #2: No

Reviewer #3: No

---

## [Author Response · Author response to Decision Letter 0]

31 Aug 2022

Dear reviewers and editorial staffs in PLOS ONE.

We are sincerely grateful for your thorough consideration and scrutiny of our manuscript, “Effectiveness of Ultrasound-guided Percutaneous Transhepatic Biliary Drainage to Reduce Radiation Exposure: A Single-Center Experience”, control number PONE-D-22-13247. Through the accurate comments made by the reviewers, we better understand the critical issues in this paper. We have revised the manuscript according to the Reviewer’s suggestions. We hope that our revised manuscript will be considered and accepted for publication in the PLOS ONE. We acknowledge that the scientific and clinical quality of our manuscript was improved by the scrutinizing efforts of the reviewers and editors.

---

## [Decision Letter · Decision Letter 1]

25 Oct 2022

Effectiveness of Ultrasound-guided Percutaneous Transhepatic Biliary Drainage to Reduce Radiation Exposure: A Single-Center Experience

PONE-D-22-13247R1

Dear Dr. Nam,

We’re pleased to inform you that your manuscript has been judged scientifically suitable for publication and will be formally accepted for publication once it meets all outstanding technical requirements.

Kind regards,

Thomas Karlas, M.D.

Academic Editor

PLOS ONE

Additional Editor Comments (optional):

Dear authors,

Thank you for re-submitting your manuscript after revision. The reviewers have evaluated your work and have no further comments.

I congratulate you to the successful review process.

Reviewers' comments:

Reviewer's Responses to Questions

**Comments to the Author**

1. If the authors have adequately addressed your comments raised in a previous round of review and you feel that this manuscript is now acceptable for publication, you may indicate that here to bypass the “Comments to the Author” section, enter your conflict of interest statement in the “Confidential to Editor” section, and submit your "Accept" recommendation.

Reviewer #1: All comments have been addressed

Reviewer #2: (No Response)

Reviewer #3: All comments have been addressed

2. Is the manuscript technically sound, and do the data support the conclusions?

Reviewer #1: Yes

Reviewer #2: Yes

Reviewer #3: Yes

3. Has the statistical analysis been performed appropriately and rigorously? 

Reviewer #1: Yes

Reviewer #2: Yes

Reviewer #3: Yes

4. Have the authors made all data underlying the findings in their manuscript fully available?

Reviewer #1: Yes

Reviewer #2: Yes

Reviewer #3: Yes

5. Is the manuscript presented in an intelligible fashion and written in standard English?

Reviewer #1: Yes

Reviewer #2: Yes

Reviewer #3: Yes

6. Review Comments to the Author

Reviewer #1: The authors completely responded all my concerns. I have no further comments. Thank you for your efforts.

Reviewer #2: The authors answered my questions quite adequately. However, this descriptive "case series" is still not inducing a high level of enthusiasm. If you decide to accept the MS I would only request to tune down the statement that PTBD should be better than ERCP for treating bile leaks in the MS.

Reviewer #3: (No Response)

7. PLOS authors have the option to publish the peer review history of their article (what does this mean?). If published, this will include your full peer review and any attached files.

Reviewer #1: No

Reviewer #2: No

Reviewer #3: No

---

## [Editor Report · Acceptance letter]

27 Oct 2022

PONE-D-22-13247R1 

Effectiveness of Ultrasound-guided Percutaneous Transhepatic Biliary Drainage to Reduce Radiation Exposure: A Single-Center Experience 

Dear Dr. Nam:

I'm pleased to inform you that your manuscript has been deemed suitable for publication in PLOS ONE. Congratulations! Your manuscript is now with our production department. 

Kind regards, 

on behalf of

Dr. Thomas Karlas 

Academic Editor

PLOS ONE